# The Transcription Factor *MiMYB8* Suppresses Peel Coloration in Postharvest ‘Guifei’ Mango in Response to High Concentration of Exogenous Ethylene by Negatively Modulating *MiPAL1*

**DOI:** 10.3390/ijms25094841

**Published:** 2024-04-29

**Authors:** Muhammad Muzammal Aslam, Mingrui Kou, Yaqi Dou, Shicheng Zou, Rui Li, Wen Li, Yuanzhi Shao

**Affiliations:** 1Sanya Nanfan Research Institute, Hainan University, Sanya 572025, China; muzamal.aslam@yahoo.com (M.M.A.); leisurejy@outlook.com (M.K.); hnudyq@outlook.com (Y.D.); 996420@hainanu.edu.cn (S.Z.); lirui@hainanu.edu.cn (R.L.); 2School of Tropical Agriculture and Forestry, Hainan University, Haikou 570228, China

**Keywords:** mango fruit, exogenous ethylene, anthocyanin synthesis, MYB TFs, fruit ripening

## Abstract

Anthocyanin accumulation is regulated by specific genes during fruit ripening. Currently, peel coloration of mango fruit in response to exogenous ethylene and the underlying molecular mechanism remain largely unknown. The role of MiMYB8 on suppressing peel coloration in postharvest ‘Guifei’ mango was investigated by physiology detection, RNA-seq, qRT-PCR, bioinformatics analysis, yeast one-hybrid, dual-luciferase reporter assay, and transient overexpression. Results showed that compared with the control, low concentration of exogenous ethylene (ETH, 500 mg·L^−1^) significantly promoted peel coloration of mango fruit (cv. Guifei). However, a higher concentration of ETH (1000 mg·L^−1^) suppressed color transformation, which is associated with higher chlorophyll content, lower a* value, anthocyanin content, and phenylalanine ammonia-lyase (PAL) activity of mango fruit. M. indica myeloblastosis8 *MiMYB8* and *MiPAL1* were differentially expressed during storage. MiMYB8 was highly similar to those found in other plant species related to anthocyanin biosynthesis and was located in the nucleus. MiMYB8 suppressed the transcription of *MiPAL1* by binding directly to its promoter. Transient overexpression of MiMYB8 in tobacco leaves and mango fruit inhibited anthocyanin accumulation by decreasing PAL activity and down-regulating the gene expression. Our observations suggest that MiMYB8 may act as repressor of anthocyanin synthesis by negatively modulating the *MiPAL* gene during ripening of mango fruit, which provides us with a theoretical basis for the scientific use of exogenous ethylene in practice.

## 1. Introduction

Anthocyanins, a type of flavonoid, are present in a variety of plants and provide numerous benefits to plant life. Additionally, anthocyanins facilitate pollination and reproduction, and aid in protecting plants from environmental stress [1,2]. The enzyme cascade beginning with phenylalanine leads to the formation of anthocyanins. The process of anthocyanin biosynthesis is mediated by several enzymes including chalcone synthase (CHS), UDP-glucose/flavonoid 3-O-glucosyltransferase (UFGT), phenylalanine ammonia-lyase (PAL), flavanone 3-hydroxylase (F3H), dihydroflavonol 4-reductase (DFR), and anthocyanin synthase, which work together to produce these compounds [3,4].

Transcription factors play a crucial role in regulating anthocyanin biosynthesis [5]. In eggplant, a COP1 interactive protein positively regulates anthocyanin accumulation and fruit size [6]. The *V-Myb* myeloblastosis viral oncogene homolog (MYB) transcription factors (TFs) family is extensively abundant in plant species, which is responsible for regulating the expression of a series of structural genes [7]. Upon detailed analysis, 22 subgroups of R2R3-MYB have been identified, and each exhibits distinct characteristics [8,9]. An increasing number of studies indicate that MYB transcription factors are involved in plant anthocyanin synthesis; for example, homologs of *MdMYB1* have been identified in apples and are known to be major activators of anthocyanin synthesis [10]. *CmMYB21* is identified to act as a repressor of anthocyanin synthesis, thus inhibiting the production of pigments in chrysanthemum [11].

*V-Myb* myeloblastosis viral oncogene homolog (MYB) transcription factors (TFs) control the expression of anthocyanin biosynthesis in response to different stimuli, which may be internal or external. Higher temperatures and nitrate levels suppress the expression of the activators; while exposure to light, lower temperatures, jasmonic acid, and ethylene induce its expression, thus promoting anthocyanin synthesis [12,13]. Research has shown that when ethylene is applied, the expression of *PpMYB10* and *PpMYB114*—activators of the R2R3-MYB—is decreased, and the expression of the R2R3-MYB repressor such as *PpMYB140* is increased, leading to reduced synthesis of anthocyanins in pear fruit [14]. Figueroa et al. (2021) [15] reported that ethylene application at the immature stage of *Fragaria chiloensis* fruit represses anthocyanin biosynthesis with a concomitant accumulation of lignin. This suggests that the effect of ethylene in regulating anthocyanin biosynthesis may vary significantly between different species, warranting further investigation.

In recent years, an increasing number of MYB transcription factors have been identified, which activate genes related to anthocyanin biosynthesis [16,17]. The transcriptional levels of *VvMYBA6*, *VvMYBA7*, *VvMYBA1*, and *VvMYBA5* were analyzed in the grape variety ‘Yan’, and a positive correlation with anthocyanin synthesis was discovered [18]. It was observed that the overexpression of *VviMyb4*-like in tobacco plants had an inhibitory effect on anthocyanin biosynthesis [19]. Expression of *VviMyb4a* and *VviMyb4b* in grapevine demonstrated their ability to repress genes related to the general phenylpropanoid pathway, while *VviMybC2-L* and *VviMybC2-L3* resulted in decreased synthesis of anthocyanins [20]. The MDMYB1 transcription factor is involved in the increased production of anthocyanin in apple [21]. Nevertheless, the exact role of MYB transcription factors in the production of anthocyanins in mango fruit in response to exogenous ethylene remains largely unknown.

Mango (*Mangifera indica* L.) is one of the most delicious tropical fruits and is categorized as a climacteric fruit. Mangoes ripen quickly during storage at ambient temperature and have a short shelf life [22,23]. Ethylene plays important roles in initiating and coordinating diverse processes such as color development, softening, and aroma formation [24]. The purpose of the current research was to explore the potential role of MYB transcription factor in the regulation of anthocyanin biosynthesis in response to exogenous ethylene. In this study, we examine the effects of different concentrations of exogenous ethylene on peel color change and pigment contents in ‘Guifei’ mango fruit, and look into the potential role of *MiMYB8* transcription factor (TFs), as well as interaction with *MiPAL1* in the regulation of anthocyanin biosynthesis. This research explored the possibility that the repressor transcription factor may be involved as mediators of anthocyanin biosynthesis.

## 2. Results

### 2.1. The Effect of Exogenous Ethylene (ETH) on Peel Coloration in Mango Fruit

The effects of varying concentrations of ETH on the peel color of mango stored at 20 °C were studied (Figure 1). The peels of fruit treated with a low concentration of ETH (500 mg·L^−1^) displayed a notable change in color after 12 d and 18 d. However, the green color of fruits treated with a higher concentration of ETH (1000 mg·L^−1^) was observed, which could be attributed to several factors including a decrease in a* value, an increase in chlorophyll content, and a decrease in anthocyanin content (Figure 1B,C,E). Additionally, there was a reduction in Phenylalanine ammonia-lyase activity (PAL) (*p* < 0.05) in fruits treated with a concentration of ETH compared to untreated control fruits (Figure 1D).

### 2.2. Transcriptomic Expression Analysis of MiPAL in Mango Fruit

To explore the changes in the expression pattern of the Phenylalanine ammonia-lyase activity (PAL) gene in mango fruit during storage, we analyzed RNA-seq data and performed qRT-PCR analysis. Venn diagram and heat map depicted 9 PAL genes that were differentially expressed based on the FPKM (Fragments per Kilobase Million) data. The RNA sequencing data revealed that the expression levels of both PAL1 and PAL2 genes were higher in the control group compared to the treatment group. The down-regulation of expression was mainly prominent on the 18th day (Figure 2A,B). The results of qRT-PCR also showed that *MiPAL1* was significantly down-regulated after 18 d, which coincided with the results of RNA-seq data (Figure 2C). *MiPAL2* was also down-regulated, but no obvious difference was observed among different treatments on day 18 (Figure 2D), indicating that *MiPAL1* plays a crucial role in suppressing peel coloration in the ripening of mango fruit.

### 2.3. Transcriptomic Expression Analysis of MiMYB in Mango Fruit

Figure 3A,B exhibits the expression differences of MYB genes between the control group and treatments. A total of 26 DEGs were identified across all three comparison groups (Figure 3B). A histogram showed that the CK6d vs. H-ETH6d comparison group had the highest count of down-regulated and up-regulated MYB genes, with 45 and 36, respectively (Figure 3B).

The qRT-PCR analysis revealed a significant decrease in expression levels of *MiMYB5*, *MiMYB6*, and *MiMYB8* after 18 d of ETH treatment (Figure 3C–E). Notably, the down-regulation was most prominent in *MiMYB8*, which is consistent with the results from RNA-seq analysis, and this further emphasizes the involvement of the MYB gene in response to exogenous ethylene in mango fruit.

### 2.4. Phylogenetic Analysis and Subcellular Localization of MiMYB8 Gene

To construct a phylogenetic analysis, we selected the sequences, particularly for MYB genes. In the case of Arabidopsis, known for its abundance of MYB genes, we arranged sequences based on their functional distinctions, expression patterns, and potential interactions with other genes or proteins. Furthermore, we integrate sequences from closely related species to enrich the evolutionary context and reduce the redundancy. The phylogenetic tree constructed from *MiMYB8* from mango, as well as *AtMyb61*, *AtMYB58*, *AtMYB52*, *AtMYB3*, *AtMYB90*, *AtMYB14*, *AtMYB111*, *AtMYB78*, *VvMYB60*, *VvMYB17*, *SmMYB2*, *MdMYB30*, and *MdMYB36* from other species, revealed that most of the Arabidopsis thaliana MYBs clustered together. Furthermore, *MiMYB8* from mango was also found within this cluster. This confirmed that *MiMYB8* belonged to the R2R3-MYB family, as previously assumed, and suggests that *MiMYB8* has a similar function, likely due to the analogous conserved domains present among these MYBs (Figure 4A).

To investigate the expression site of *MiMYB* in mango, the C-terminal regions of *MiMYB8* were fused with green fluorescence protein (GFP) and then expressed transiently in the leaf of *Nicotiana benthamiana* epidermal cells under the control of promoter CaMV 35S (Figure 4B). The GFP signal of the control (pCAMBIA1302 GFP) was perceived in both the nucleus and cytoplasm, while the GFP signals of pCAMBIA1302-*MiMYB8* GFP were present exclusively in the nuclei (Figure 4C). This demonstrated that *MiMYB8* acts as a nuclear protein, playing a dominant role in the anthocyanin biosynthesis of mango.

### 2.5. Interaction of MiMYB8 and MiPAL1 Genes

A dual luciferase assay was conducted to investigate the effect of *MiMYB8* on *MiPAL1* promoter activity in vivo using tobacco (*N. benthamiana*) leaves. Initially, we analyzed the promoter region of *MiPAL1* and *MiMYB8* genes in postharvest mangoes. Within the 1513 bp *MiPAL1* promoter region, seven MYB binding sites were identified, including MYC, chs-CMA2a, and MBS motifs (Figure 5C). Subsequently, we assessed the ability of the *MiPAL1* promoter to drive gene expression in tobacco cells. For this purpose, the pGreenII 0800-Luc vector containing the *MiPAL1* promoter sequence was transiently transformed into tobacco leaves alongside a control vector (pGreenII 62) and monitored for luciferase activity. The results revealed a strong expression of luciferase, indicating the effective functionality of the *MiPAL1* promoter in driving gene expression in tobacco cells. Following the evaluation of *MiPAL1* promoter activity, we investigated the impact of *MiMYB8* expression on *MiPAL1* promoter function. The 35S-*MiMYB8* vector, comprising the *MiMYB8* gene, was co-transformed into tobacco leaves along with the pGreenII 0800-Luc vector containing the *MiPAL1* promoter sequence and the control vector. Remarkably, the co-expression of *MiMYB8* resulted in a significantly lower relative ratio of firefly (LUC) to Renilla (REN) luciferase compared to the control (Figure 5A,B). This observation suggests that *MiMYB8* specifically binds to the MiPAL1 promoter and effectively suppresses the activity of the *MiPAL1* gene, reducing its ability to drive gene expression in tobacco cells.

In Y1H analysis, the transformed yeast Y187 was found to grow normally on SD/-Trp/-His medium, pHIS2-*MiPAL1*pro. However, no growth was observed when 10 mM 3-AT was added. This result suggests that 3-aminotriazole (3-AT) at a concentration of 10 mM can inhibit the self-activation of the *MiPAL1* promoter. Furthermore, co-transformed yeast cells with pGADT7-53 and pHIS2 (a negative control), pHIS2-*MiPAL1*pro and pGADT7-*MiMYB8* constructs also grew well on the medium (Trp/SD/Leu). Moreover, Y187 yeast cells co-transformed with pGADT7-*MiMYB8* constructs and pHIS2-*MiPAL1*pro were still able to grow normally on 10 mM 3-AT SD/-Leu/-Trp/His medium (Figure 5D,E). These results indicate that the *MiMYB8* protein is capable of binding to the *MiPAL1* promoter.

### 2.6. Transient Overexpression (OE) of MiMYB8 in Tobacco Leaf

Agro-infiltration of the *Nicotiana benthamiana* plant was used to investigate the function of *MiMYB8* on anthocyanin synthesis, wherein a 35S-promoter-driven construct of this gene was individually inserted into the leaves of the plants. The result demonstrated that an expression level for *MiMYB8* was significantly increased on the 3rd and 6th day post-injection (Figure 6A); however, overexpression of *MiMYB8* decreased both expression levels and phenylalanine ammonia-lyase activity of *MiPAL1* compared with the control (Figure 6B,C).

### 2.7. Transient Overexpression (OE) of MiMYB8 in Mango Fruit

The results of overexpressing *MiMYB8* on *MiPAL1* expression levels, PAL activity, and anthocyanin content of mango flesh discs on 3 d and 6 d post-treatment were seen and compared in (Figure 7). The control fruit discs were significantly darker in color, indicating higher anthocyanin contents, than those of the overexpressed *MiMYB8* disc (Figure 7A). In addition, OE-*MiMYB8* treatment down-regulated the expression of the *MiPAL1* gene (Figure 7B), and there was a significant decrease in both PAL enzyme activity and anthocyanin content in OE-*MiMYB8* fruit compared to the control fruits after 3 d and 6 d, especially on day 3 (Figure 7D,E). The overexpression of *MiMYB8* in mango resulted in a noticeable decrease in anthocyanin content, leading to lighter coloration compared to the control. This reduction was accompanied by a down-regulation of the *MiPAL1* gene expression, suggesting that *MiMYB8* may regulate genes involved in anthocyanin biosynthesis. Additionally, the decrease in PAL enzyme activity further supports the idea that *MiMYB8* negatively influences the anthocyanin biosynthetic pathway. These findings highlight the significant role of *MiMYB8* in controlling anthocyanin accumulation in mango fruit and suggest that manipulating its expression levels could be a promising strategy for altering fruit pigmentation traits. Thus, it appears that *MiMYB8* is involved in the inhibition of anthocyanin biosynthesis under exposure to higher levels of exogenous ethylene in mango fruit.

## 3. Discussion

Mango skin color is an essential trait for marketability and aesthetics. As the color of the peel impacts consumer preferences, the accumulation of anthocyanin within the skin of mangoes has been found to improve quality, pathogen resistance, and cold tolerance [25]. Mangoes treated with ETH may appear to have a normal peel color; however, the quality of the mangoes can be adversely affected due to an excessive level of maturity [26]. MYB proteins play a critical role in modulating anthocyanin biosynthesis, acting both as activators and inhibitors of this process. Members of this family have been found in some horticultural plant species that positively control temporal and spatial production of anthocyanins through the phenyl-propane metabolic pathway [27]. However, recent studies have highlighted that there is increasing evidence that they may also exert negative regulation [28,29]. In the present study, the transcription factor, *MiMYB8* was identified as a repressor of anthocyanin biosynthesis in the ‘Guifei’ mango, confirming its negative regulatory role.

Ethylene (C_2_H_4_) is a gaseous natural plant hormone that can have both beneficial and detrimental effects on fruit ripening [30,31]. In this study, the lower concentration of ethylene treatment resulted in a relatively quicker decrease in chlorophyll content and a slight rise in PAL activity (Figure 1). Transcriptomic analysis led to the identification of two genes, *MiPAL1* and *MiPAL2*, which are closely linked to the biosynthesis of anthocyanins (Figure 2). These findings are consistent with previous studies investigating the function of *CsPAL4*, which is a gene highly expressed in buds and young leaves and involved in the biosynthesis of anthocyanins in purple-leaf tea and *Arabidopsis* [32]. Analysis of RNA sequencing has revealed several *MiMYB* transcription factors (*MiMYB5*, *MiMYB6*, and *MiMYB8*) that can boost anthocyanin biosynthesis through upregulation of the gene encoding phenylalanine ammonia-lyase (PAL). Previous studies have identified *VvMYB15* as a transcription factor that plays a role in activating genes (*VvPAL3 DcPAL3)* associated with anthocyanin biosynthesis in grapevines and carrots, specifically by increasing the expression of PAL enzyme [33,34]; our results are consistent with these findings.

The qRT-PCR analysis revealed that the expression of three key anthocyanin repressors *MiMYB5*, *MiMYB6* and *MiMYB8* had a high correlation with accumulation of anthocyanin (Figure 3). This confirms similar results from other studies into repressors of anthocyanin production, such as *FaMYB1* in strawberries and *PpMYB18* in peaches, which were both found to be highly expressed in developmental stages and anthocyanin-rich tissues [35]. These findings suggest that there is a feedback loop involved in the regulation of anthocyanin biosynthesis. *MdMYB16* is a MYB repressor found in apples that appears to show expression levels that are negatively correlated with anthocyanin biosynthetic genes, anthocyanin accumulation, and other positive regulators. This effect is likely caused by its ability to bind to and repress the function of its regulatory motif, thereby directly inhibiting anthocyanin biosynthesis [36]. MYB repressors are activated by activators, thereby generating negative feedback.

To verify the function of *MiMYB8* in anthocyanin biosynthesis of mango fruit, subcellular localization of pCAMBIA1302-*MiMYB8*-GFP was detected and found to be localized in the nucleus (Figure 4), indicating that *MiMYB8* is nuclear proteins. To further confirm that *MIMYB8* interact with *MIPAL1* in vivo, dual luciferase and YIH assays were conducted (Figure 5). The results demonstrated that *MiMYB8* inhibited anthocyanin biosynthesis by binding to *MIPAL1* and disrupting the MYB-PAL complex, thereby disrupting the transcription of the anthocyanin biosynthetic gene. The findings of Yang et al. (2021) [37] regarding the subcellular localization of a protein are echoed in the current research, which suggests that the functional analogy between these two proteins could lead to pigmentation patterns.

The overexpression of the *MiMYB8* gene in tobacco and mango plants was studied to confirm their roles in anthocyanin biosynthesis (Figure 6 and Figure 7). Results revealed that the expression level of *MiMYB8* was significantly increased; however, the expression of *MiPAL1*, PAL activity, and anthocyanin content decreased in both tobacco leaf and mango discs. Thus, these results proposed that *MiMYB8* acts as an inhibitory regulator of anthocyanin biosynthesis, which is consistent with genes such as *SlMYB-ATV* in tomatoes [38], *PtMYB182* in poplar [39], and *VvMYBC2-L1* in grapevine [40] that act as negative regulators of both anthocyanin and phenylpropanoid accumulation. Overall, our results demonstrated that the *MiMYB8* transcription factor plays a role in the inhibition of anthocyanin biosynthesis under exposure to higher levels of exogenous ethylene in mango fruit.

## 4. Materials and Methods

### 4.1. Plant Materials and Treatment

Commercially matured (around 110 d after the full-bloom stage) mango (*Mangifera indica* L.) fruits were carefully and manually harvested from Lingshui County, specifically in Yingzhou town, which is located at coordinates (108°37′–111°03′ W and 18°10′–20°10′ N), in Hainan, China. Throughout the growth period of mangoes in Yingzhou, spanning from December to April, the average temperature maintains around 22 °C, and is accompanied by approximately 88% relative humidity. These freshly harvested fruits were transported to the laboratory at ambient temperature, ranging between 20 °C and 25 °C, within 3 h. A total of 500 fruits, similar in size and without any external damage, were soaked in a (0.1%) sodium hypochlorite solution, and dried at room temperature (25 °C, RH 90%). To elucidate the color response to exogenous ethylene, we selected concentrations of 500 and 1000 mg·L^−1^ of ethephon, which is commonly employed in practice. These mango fruits were grouped into three lots randomly (150 mango fruit for each group) and immersed in (pure water as control), 500 and 1000 mg·L^−1^ ethephon solution for 10 min, respectively. After allowing the fruits to dehydrate naturally at room temperature, they were carefully enclosed within unsealed polyethylene bags measuring 0.02 mm in thickness (brand: Xingfengr^®^, origin: China, dimensions: 20 × 30 cm). Subsequently, the packaged fruits were placed inside a controlled climatic cabinet set to maintain a temperature of 20 °C and a relative humidity of 90% for duration of 18 d. Afterwards, each group was divided into two parts: one part (30 fruit) was used for monitoring the peel color change of mango fruit, and the other part (about 120 fruit) was used for the measurement of other indicators. Three groups of fruits were sampled at 3 days interval. Mango flesh and peel were separated at each sampling time point, and then frozen in liquid nitrogen, ground into powder, and kept at −80 °C for further use. Each experiment consisted of three biological replicates with three uniformly sized fruits.

### 4.2. Measurement of Peel Color Characteristics of Mango Fruit

Using the Wang et al. (2023) [28] method, peel color measurements were conducted using a colorimeter (CR-10PLUS, Osaka, Japan) on four randomly selected equatorial sites across each mango’s skin. Two measurements were taken on the shady side, and the average was calculated for the following analysis. The raw data obtained comprised values for a* (indicating redness or greenness) and b* (representing yellowness or blueness). The Chrome value (C*) was determined using the formula C* = a*^2 + b*^2. The chroma a* value indicating luminosity ranging from 0 (black) to 100 (white). The chroma b* value indicating luminosity ranging from 0 (green) to 100 (red).

The chlorophyll content was measured according to the method of Chen et al. (2022) [41]. A 0.5 g sample of frozen peel powder cooled to 4 °C was extracted in the dark with an 80% acetone solution. This mixture was centrifuged three times at 12,000× *g* for 30 min; afterwards, absorbance was measured for the solution at three different wavelengths (470, 645, and 663 nm) using a UV752N spectrophotometer (YoKe Instrument Co., Ltd., Shanghai, China). The extracted chlorophyll content was finally calculated from the absorbance measurements.

For determination of anthocyanin content, a total of 0.5 g of frozen peel powder cooled to 4 °C was put into an 8 mL tube containing a 1% hydrochloric acid-methanol solution. The extraction procedure was conducted at 4 °C for 24 h. After this period, the solution was centrifuged for 15 min and 3 mL of the extract was generated. The extract was then transferred into the light cuvette, whose diameter should be 1 cm and the values of absorbance were measured at 530 nm and 600 nm. To ensure accuracy, the procedure was repeated three times, with separate hydrochloric acid (1%) and methanol solutions used as a blank control for each repetition. This method was adopted by Zhang et al. (2001) [42] for determining the anthocyanin content in the samples.

### 4.3. Determination of PAL Enzyme Activity

The reaction mixture for measuring the PAL activity was prepared by mixing of peel tissue sample (0.5 g) with 5 mL of pre-cooled pH 8.8 buffer (at 4 °C), and Borate buffer (3 mL, 50 mmol L^−1^, and pH 8.8) with 500 μL of crude enzyme. This mixture was homogenized and then centrifuged at 12,000× *g* for 20 min at 4 °C. After this, L-phenylalanine (0.5 mL, 20 mmol L^−1^) was added to the supernatant of the mixture and the centrifuge tube was incubated at 37 °C for 10 min for further analysis. The reaction was further incubated for 1 h and then terminated with trichloroacetic acid (0.6 mL per 100 mL). The activity of PAL was measured at 290 nm (A290), with an optical density increase of 2.9 × 10^−7^ (0.01 OD) [43].

### 4.4. Analysis of Differentially Expressed Genes (DEGs)

Individual libraries were prepared with barcodes and pooled for commercial sequencing on an Illumina HiSeq2500 platform by Gene Denovo Biotechnology Co. (Guangzhou, China). An FPKM (fragment per kilobase of transcript per million mapped reads) value was calculated for each transcript to quantify its relative abundance using StringTie software (version 2020). FPKM values were used to identify differentially expressed genes (DEGs) between the control and treated mango fruit. The genes with the parameter of false discovery rate (FDR) below 0.05 and absolute fold change ≥ 2 were considered differentially expressed genes [44].

### 4.5. RNA Extraction and qRT-PCR Analysis

An improved CTAB method established in our laboratory was employed to isolate total RNA from frozen peel samples, assessed for degradation or contamination via 1% agarose gel electrophoresis, and then used to synthesize cDNA [45]. HiScript Prime Script™ (Nanjing, China’s), 1st Strand cDNA Synthesis Kit can be utilized. RT-qPCR was then executed on the cDNA using (SYBR) Premix Ex Taq (HiScript R, Nanjing, China), according to the provided procedure. Quantitative RT-PCR was done using qPCR SuperMix and CFX96 Real-Time PCR system, with primers designed by Primer Explorer V5 tool (https://primerexplorer.jp/e/) URL (accessed on 20 March 2022). The analysis of RT-PCR was conducted on *MiActin* with CFX96 real-time PCR system. The primers were organized through Taihe Biotech Co. (Shanghai, China), which is mentioned in (Appendix A). The protocol included 95 °C for 5 min (denaturation) → 95 °C for 5 s (denaturation) → 60 °C for 30 s (annealing) → 72 °C for 10 s (extension) X 40 cycles. Relative abundances were calculated using the 2^−ΔΔCt^ method that was declared by Livak and Schmittgen (2001) [46].

### 4.6. Gene Cloning and Promoter Analysis

We can use the DNA primers listed in Appendix A to amplify specific sequences in the DNA. The transcriptomic database was explored for sequences of *MiPAL* and *MiMYB* flavonoid-related genes. The PCR-amplified coding sequences were then translated and sequenced using NCBI Open Reading Frame Finder and DNAMAN software (version 12.1). The coding sequences of *MiMYB8* (Cluster-23630.11139_1R) are being examined. The *MiPAL1* promoter sequence was analyzed for putative cis-acting elements through an online database (http://bioinformatics.psb.ugent.be/webtools/plantcare/html/ URL (accessed on 25 October 2021).

### 4.7. Subcellular Localization Analysis

Five-week-old tobacco (*Nicotiana tabacum*) plant leaves were agro-infiltrated by using *Agrobacterium tumefaciens* GV3101 containing recombinant vector pCAMBIA1302-*MiMYB8*-GFP, generated by amplifying the coding region of *MiMYB8* cDNA, respectively, using polymerase chain reaction (PCR) lacking stop codon, and ligating them into the vector pCAMBIA1302-GFP. Alongside, a control vector was also infiltrated. A total of 48 h after transformation, the *A. tumefaciens* lines were suspended at an optical density of 1.0 in the infiltration buffer. To observe the fluorescence of green fluorescent protein (GFP), a Leica DM6000B Fluorescence Microscope (Leica, Hong Kong, China) was used, along with DAPI (Real Times, Hong Kong, China) to confirm nuclear localization.

### 4.8. MiPAL1 Promoter Amplification and Yeast One-Hybrid (Y1H) Assay

Y1H assay was conducted on the Y187 strain cells, following the manufacturer’s protocols. In the experiment, the coding sequences (CDS) for the three MYB genes, *MiMYB5*, *MiMYB6*, and *MiMYB8*, were implanted into the vector (pGADT7) to construct the recombinant plasmids with labels *MiMYB5*-AD, *MiMYB6*-AD, and *MiMYB8*-AD. Y187 cells were co-transformed with recombinant plasmids pHIS2-PAL: *MiMYB5*-AD, pHIS2-PAL: *MiMYB6*-AD, and pHIS2-PAL: *MiMYB8*-AD to measure the interactions between *MiMYB5*, *MiMYB6*, and *MiMYB8* genes and the *MiPAL1* promoter. Media that lacked Leu, Trp, and His, and were added with 3AT (Trp/SD/Leu/His/3AT) were used to analyze the interactions. The sequences of the *MiPAL1* promoter used in the pHIS2 vector to generate pHIS2-PAL can be found in (Appendix A).

### 4.9. Dual-Luciferase Reporter (DLR) Assays

Genomic DNA was used to amplify the *MiPAL1* promoter (−1443 bp upstream from the ATG start site), and *MiMYB8* coding region, which was then inserted into the respective vectors pGreenII-0800 and pGreenII 62-SK. This generated the *MiPAL1*pro-Luc reporter plasmid and 35S-*MiMYB8* effector plasmids. Plasmids containing transcription factor and negative control pGreenII-62SK plasmids were transformed into *Agrobacterium tumefaciens* (GV310) and introduced into the leaves of tobacco (3 weeks old). After two days, the effect of the transcription factor on the promoter was measured using a kit (Dual-Luciferase^®^ Reporter Assay). The REN and LUC activities were utilized to determine the degree of promoter activation, with biological replicates for each transcription factor-promoter interaction.

### 4.10. Transient Overexpression (OE) of MiMYB8 in Tobacco Leaf

For transient overexpression, we used the method described by Liu et al. (2021) [47] with slight modifications. We cloned the CDS sequence of *MiMYB8* and inserted it into the overexpression vector pGreen II 62-SK to create the fusion vector SK-*MiMYB8*. Primers for transient overexpression in tobacco (*Nicotiana benthamiana*, *NB*) were designed and are listed in Appendix A.

Overnight incubation in LB medium with appropriate antibiotics (50 mg mL^−1^ of kanamycin and 25 μL.mL^−1^ of rifampicin) was employed to transform two constructs into Agrobacterium tumefaciens GV3101. The cultures were subsequently adjusted to an optical density (OD_600_ = 0.75), employing an infiltration buffer composed of 10 mM MES, 10 mM MgCl_2_, and 150 mM acetosyringone, and adjusted to a pH of 5.6. Following this adjustment, the cultures were permitted to incubate at room temperature (28 °C) for 2–3 d.

After preparing the suspensions, 500 mL of Agrobacterium tumefaciens GV3101 suspension (designated Empty) and Agrobacterium tumefaciens GV3101 carrying 35S-*MiMYB8* suspension (designated OE-*MiMYB8*), were injected into the leaf’s back using a needleless syringe with gentle rubbing, respectively. The tobacco plants were then placed in a lighted growth chamber with a temperature of 22 ± 0.5 °C and a relative humidity of 60–65%. This phase was characterized by a light/dark cycle of 16:8 h, and it spanned duration of 1 d. Subsequently, samples were collected after 3 d and 6 d. Each group consisted of three replicates with 10 strains in each replicate. PAL activity and transcription level in tobacco were assessed as the same as in Section 4.10.

### 4.11. Transient Overexpression (OE) of MiMYB8 Gene in Mango Fruit

Commercially ripe ‘Guifei’ mangoes were harvested and divided into two groups, each containing 30 fruits. The peels of mangoes were cut into discs with a diameter of 6 mm and a thickness of 1 mm. These mango discs were then soaked in equilibrated suspensions for 2–3 h, placed on a non-resistant solid medium (50 mM acetosyringone) and sealed with a plastic wrap to cultivate in the dark at 25 °C for 6 d. Samples were taken every 3 d twice. The content of anthocyanin and the activity of PAL were subsequently determined, and *MiPAL1* and *MiMYB8* expression were assessed.

### 4.12. Statistical Analyses

The Sigma Stat (version 4.0) was used to analyze the Triplicate assay data using a one-way ANOVA test. Duncan’s Multiple Range Test was used to compare the means and standard errors (SEs) of the data and determine if any significant differences existed between the groups. A *p*-value of less than 0.05 indicates that there is a statistically significant difference between the compared groups or data sets.

## 5. Conclusions

The results show that ethylene at low concentration was effective in promoting the coloration of mango peel, whereas the high ethylene concentration had an inhibitory effect on the coloring process; as evidenced by a lower chroma a* value, increased chlorophyll content, reduced anthocyanin content and decreased PAL activity in comparison to the control. The *MiMYB8,* acting as a nuclear protein, could bind to the *MiPAL1* promoter and significantly suppress its transcription to inhibit anthocyanin accumulation and peel coloration during ripening of mango fruit. Consequently, we conclude that *MiMYB8* is a key negative regulator of anthocyanin biosynthesis in mango fruit peel. A better understanding of this process could be immensely valuable and provide researchers with insights for further exploration.

## Figures and Tables

**Figure 1 ijms-25-04841-f001:**
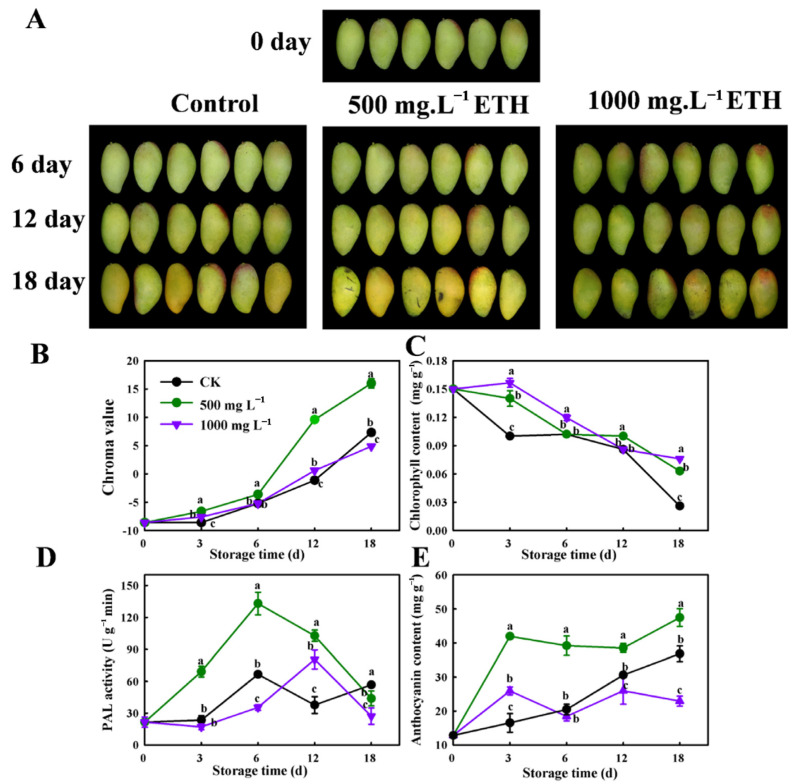
Changes in appearance and color traits related to anthocyanin biosynthesis during storage of mango fruit at 20 °C. (**A**) Peel appearance. (**B**) Chroma value. (**C**) Chlorophyll content. (**D**) Phenylalanine ammonia-lyase activity (PAL). (**E**) Anthocyanin content. Results represent the mean and standard error of three biological replicates and are statistically significant (*p* < 0.05), as indicated by lower-case letters according to Duncan’s multiple range test.

**Figure 2 ijms-25-04841-f002:**
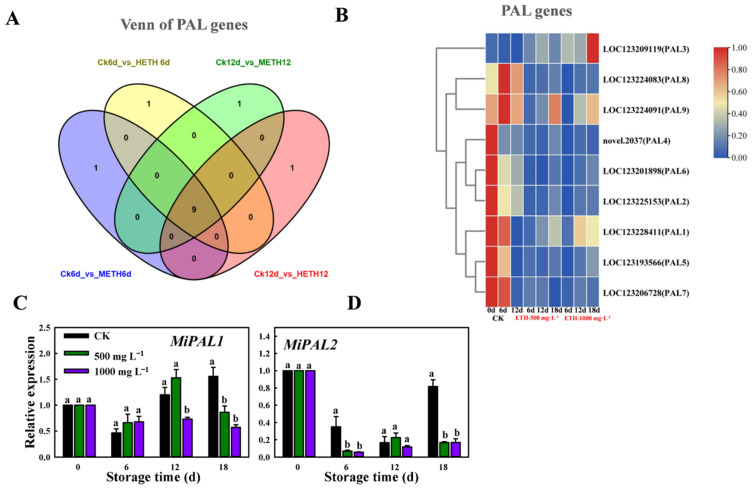
Analysis of differentially expressed genes (DEGs) and expression patterns of *MiPAL* genes during storage of mango fruit. (**A**) Venn diagram. (**B**) Heat map visualizing differentially expressed genes and FPKM data relating to MiPAL gene. (**C**) qRT-PCR analysis of *MiPAL1*. (**D**) qRT-PCR analysis of *MiPAL2* in mango fruits stored at 20 °C. Letters a and b on the bars above indicated statistically significant differences between treatments (*p* < 0.05).

**Figure 3 ijms-25-04841-f003:**
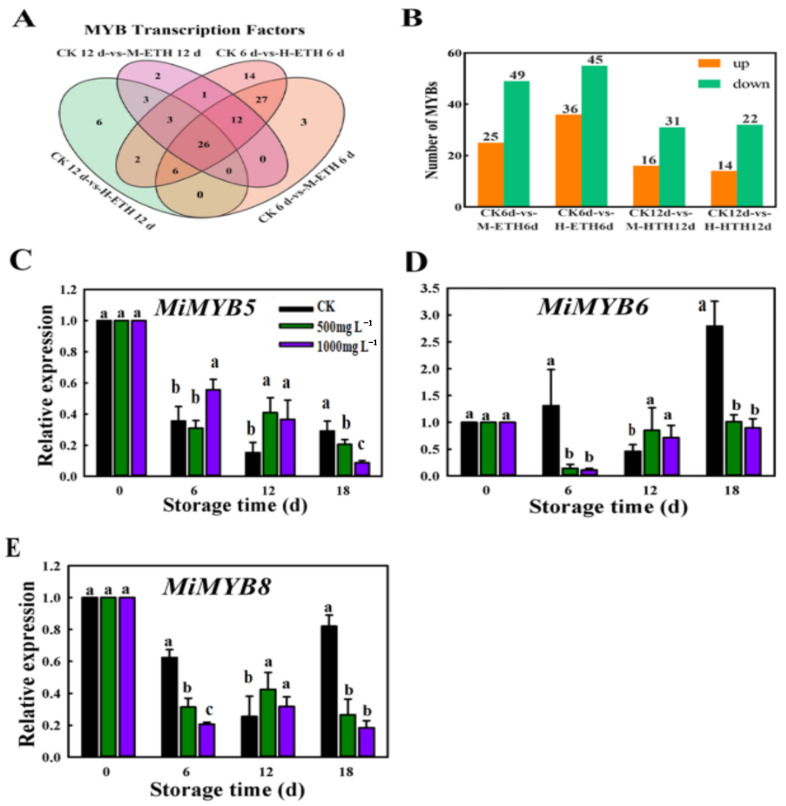
Analysis of differentially expressed genes (DEGs) and expression patterns of *MiMYB* TFs during storage of mango fruit. (**A**) Venn diagram. (**B**) A comparison of the number of up-regulated and down-regulated genes. The qRT-PCR analysis of the expression patterns of *MiMYB5* (**C**), *MiMYB6* (**D**), and *MiMYB8* (**E**). The letters a, b and c above bars indicate statistically significant differences between the respective treatments (*p* < 0.05).

**Figure 4 ijms-25-04841-f004:**
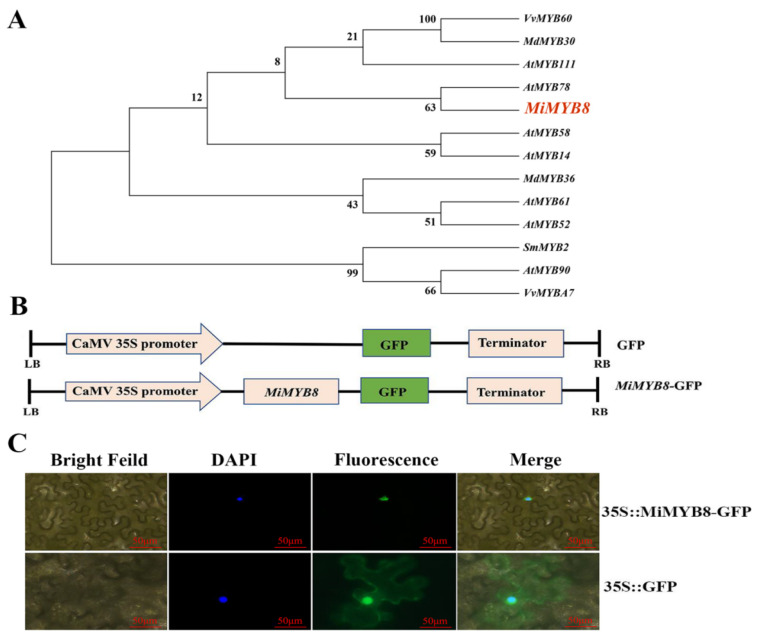
Screening, identification, and subcellular localization analysis of *MiMYB8*. (**A**) Phylogenetic tree of MiMYB8 and other MYBs. (**B**) Schematic diagrams of the vector constructs. (**C**) Subcellular localization of *MiMYB8* in epidermic cells of tobacco *(N. benthamiana*) leaves. Scale bar = 50 μm.

**Figure 5 ijms-25-04841-f005:**
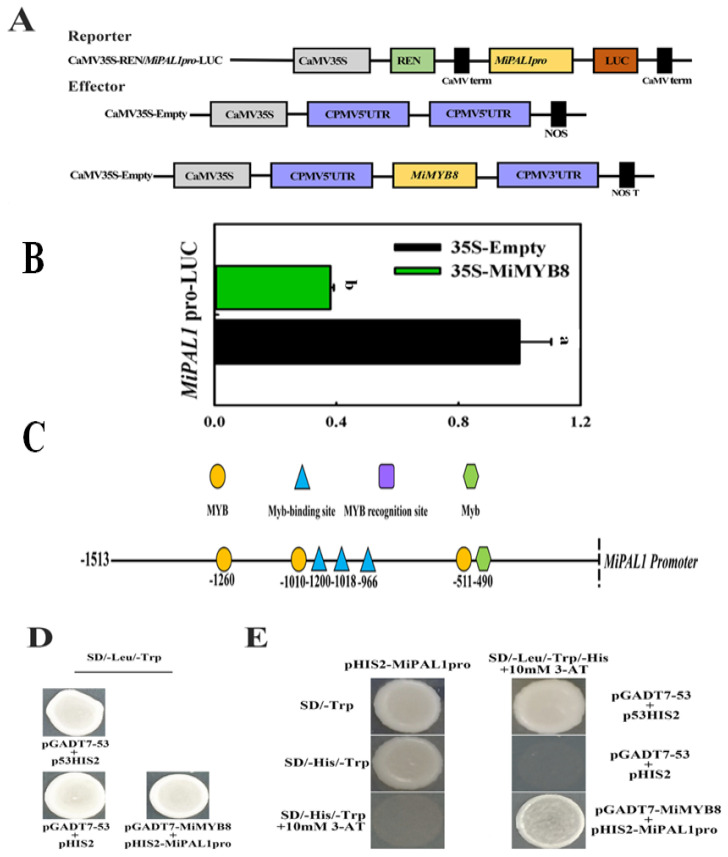
*MiMYB8* can directly bind to the *MiPAL1* promoter. (**A**) Illustration of the effector and reporter vectors used in the luciferase (Luc) transaction assay. (**B**) The ratio of LUC to REN shows the activity of *MiMYB8* on the *MiPAL1* promoter after the transformation of young Nicotiana benthamiana leaves. The values represent the mean of four biological replicates, and the vertical bars represent the standard error. (**C**) Homeopathic element analysis of *MiPAL1* promoter. (**D**) Determination of the self-activation concentration of the *MiPAL1* promoter. (**E**) Yeast one-hybrid assays between *MiMYB8* TFs and the promoter of *MiPAL1.* The letters a and b above bars indicate statistically significant differences between the respective treatments (*p* < 0.05).

**Figure 6 ijms-25-04841-f006:**
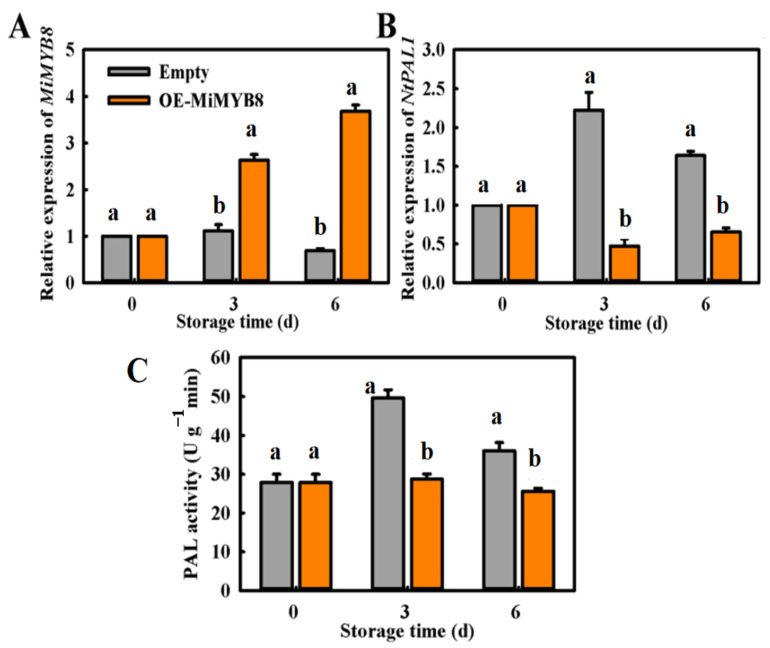
Overexpressing *MiMYB8* in *Nicotiana benthamiana* leaves suppresses *MiPAL1* expression and activity. (**A**) *MiMYB8* expression levels. (**B**) *MiPAL1* expression levels. (**C**) Phenylalanine ammonia-lyase activity in OE-*MiMYB8* leaves. The letters a and b above the bars indicate statistically significant differences for the respective treatments (*p* < 0.05).

**Figure 7 ijms-25-04841-f007:**
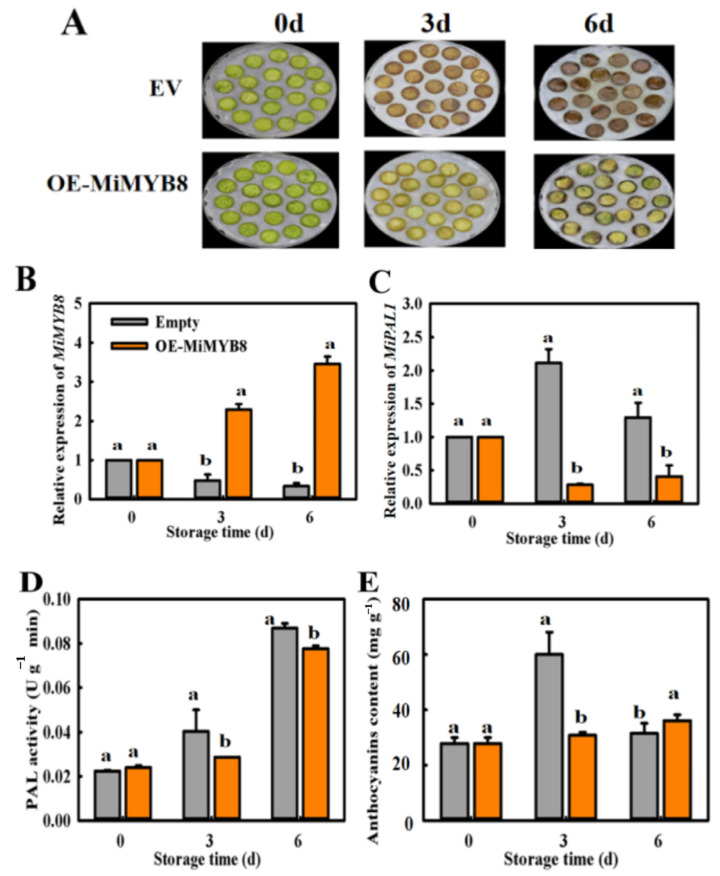
Overexpressing *MiMYB8* in mango flesh discs suppresses anthocyanin accumulation. (**A**) The appearance of mango flesh discs on days 0, 3, and 6 following the OE-*MiMYB8* gene. (**B**) *MiMYB8* expression levels. (**C**) *MiPAL1* expression levels. (**D**) Phenylalanine ammonia-lyase activity. (**E**) Anthocyanin content. The letters a and b above bars indicate statistically significant differences for the respective treatments (*p* < 0.05).

## Data Availability

Data are contained within the article and Appendix A.

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
