# Peer review of "The Transcription Factor MiMYB8 Suppresses Peel Coloration in Postharvest ‘Guifei’ Mango in Response to High Concentration of Exogenous Ethylene by Negatively Modulating MiPAL1"

_ijms, 2024, doi:10.3390/ijms25094841_

Round 1
Reviewer 1 Report
Comments and Suggestions for Authors
Here some notes, the rest on the pdf of the ms.
Fig. 1: panel A and B are not clear.
Fig. 1A: is the sense of this panel to show that 9 PAL genes are differentially expressed? Do you need a panel to say this?
Fig. 1B: it is not clear how the heat map was built. Is it based on relative FPKM data or on ratios? Moreover, I do not understand if RNAseq data (panel 1B) for PAL1 and PAL2 are in agreement with qRT-PCR data (panel 1C and 1D) for PAL1 and PAL2, respectively.
lines 106-107: MiPAL1 expression at 18H in 2A and 2B is not described.
Lines 135-141: and related fig. 4A do not make much sense unless an explanation on how the arabidopsis and other species sequences were selected. Indeed, there are many more MYB genes in plant genomes (at least 120 in arabidopsis), thus a rationale for including some and not others should be given.
Lines 159-162: this experiment could be meaningful only if the MiPLA1 promoter could drive gene expression in tobacco leaf cell. But this could happen only if tobacco trans-acting factors could transactivate the MiPAL1 promoter. As this has never been shown, Authors should at first show the ability of the MiPAL1 promoter to drive gene expression in tobacco cells, and only then show that this ability can be impaired by the expression of MiMYB8.
Methods: there are 2 variables in the described experiments: the first is the treatment (3 samples, i.e. cont, 500 mg ETH, 1000 mg ETH), the second is time (5 time points, from day 0 to 18). Statistical support on the variation among treatment has been assessed, while the one among different time points was not.

Comments on the Quality of English LanguageThe quality is not uniform along the ms. Low levels are frequent in, but not limited to, the material and methods section, which is an integral part of the ms and should also be clear.
Author Response
Dear IJMS Journal editor,
We greatly appreciate you, the Consulting Editor and the reviewers for your kind suggested revisions of our Manuscript ID: ijms-2913186. “The transcription factor MiMYB8 suppresses peel coloration in postharvest 'Guifei' mangoes in response to high concentrations of exogenous ethylene by negatively modulating MiPAL1”. We have fully considered your suggestions and exerted our efforts to revise the manuscript. We submit here the revised manuscript as well as our point-by-point responses. All changes have been marked in red in the revised manuscript, and response are listed below in blue color according to the comments.

Reviewer 2 Report
Comments and Suggestions for Authors
Dear Editor
The MS is well written, but the research question and hypothesis are not established. Also, there are some points that need to be improved in the method section. Therefore, major corrections are required in the study.
After corrections, the MS can be accepted for publication in your journal.
Best regards

Author Response

(The authors gave the same response as above.)

Reviewer 3 Report
Comments and Suggestions for Authors
The experimental article “MiMYB8 negatively modulates MiPAL1 to suppress peel coloration of postharvest 'Guifei' mango upon exposure to high concentration of ethephone” is devoted to the study of the effect of the concentration of exogenous ethylene (ethephone) on changes in peel color
and pigment content in mango fruits, as well as exploring the potential role of MYB transcription factors and their interaction with MiPAL in the regulation of anthocyanin biosynthesis. The authors found that increasing the concentration of ethephon has an inhibitory effect on the coloring process of mango peel. It was found that the content of anthocyanins decreases as the concentration of ethephon increases, and PAL also decreases compared to the control. The authors conclude that MiMYB8 is a key negative factor regulating anthocyanin biosynthesis in mango fruit peel. The authors of the article in their study use a modern, highly sensitive and accurate tool for studying the transcriptome - the RNA sequencing method. The positive side of the article is that the authors compare the results obtained in this work with the results of other researchers. To publish a manuscript, it is recommended to eliminate the comments and follow the recommendations given in the list.
Notes and recommendations:
1. Title. The title needs to be changed so that it is of interest to IJMS readers.
2. Remove the abbreviation MiMYB8 from the name or give its decoding, since it is understandable only to a narrow circle of readers and is not generally accepted among IJMS readers.
3. Abstract. Provide the abstract in accordance with the requirements of the publication, including using the same terminology (exogenous ethylene=ethephone?).
4. Results. Line 94, 103. It is necessary to provide a decoding of the abbreviation “PAL”.
5. Figure 1B and Figure 2. It is necessary to explain the abbreviation “SK”.
6. Figures 1-3. Picture caption. It is necessary to provide a link that reflects information about the method of statistical processing of the results obtained by the authors.
7. Line 105. It is necessary to decipher the abbreviation “FPKM”.
8. Lines 125-127. It is necessary to confirm by reference that “the down-regulation was most prominent in MiMYB8, consistent with the results from RNA-seq analysis and further emphasizing the involvement of the MYB gene in response to ethylene in mango fruit.”
9. Line 169. It is necessary to provide a decoding of the abbreviation “3-AT”.
10. Lines 197-206. The authors should provide a more detailed description of the mechanism of influence of “MiMYB8” on the process of biosynthesis of mango fruits.
11. Figure 7. According to Figure 7, “MiMYB8” is involved in the inhibition of anthocyanins already on harvested fruits, however, in the article, the authors say that “MiMYB8” is involved in the inhibition of anthocyanins during the ripening of mango fruits. Clarification needed.
12. Figure 7. Picture caption. Lines 211-212. It is necessary to provide a link reflecting the method of statistical processing of the results obtained.
13. Authors should pay attention that some of the illustrations need to be included in Saplimentary.
14. After all the changes, you need to work on the text of the article in such a way that its content is understandable to a wide range of readers.
Author Response
We greatly appreciate you, the Consulting Editor and the reviewers for your kind suggested revisions of our Manuscript ID: ijms-2913186. “The transcription factor MiMYB8 suppresses peel coloration in postharvest 'Guifei' mangoes in response to high concentrations of exogenous ethylene by negatively modulating MiPAL1”. We have fully considered your suggestions and exerted our efforts to revise the manuscript. We submit here the revised manuscript as well as our point-by-point responses. All changes have been marked in red in the revised manuscript, and response are listed below in blue color according to the comments.

Round 2
Reviewer 2 Report
Comments and Suggestions for Authors
Dear Authors
Among the corrections I suggested in my previous evaluation, all other suggestions, except for the suggestion in the introduction, were not implemented. The text still needs to be developed. My suggestions are shown on the PDF file.
Best regards

Author Response
Dear editor and reviewer:
Thank you very much for your kind suggestions on our manuscript ID: ijms-2913186. “The transcription factor MiMYB8 suppresses peel coloration in postharvest 'Guifei' mango in response to high concentration of exogenous ethylene by negatively modulating MiPAL1”. We have fully considered your suggestions and exerted our efforts to revise the manuscript. The detailed responses are shown below and the change points were marked in red color in the revised manuscript.
